# OpenReview forum: "Random Character-Level Perturbations Amplify LLM Jailbreak Attacks"
_TMLR — Accepted by TMLR_

### Review · Reviewer_Kkcm · 2025-12-16

**Summary Of Contributions:**

The paper studies how minimal word‑internal character perturbations (e.g., “firearms” → “firrrearms”) interact with subword tokenization to amplify or sometimes reduce jailbreak attack success on LLMs. The authors define eight simple character‑level perturbation schemes, apply them to harmful “attack targets” with and without jailbreak templates (GCG and DeepInception), and evaluate attack success rates across four open‑source LLMs using two safety‑judge models. They then conduct a series of analyses on tokenization fragmentation, representation drift, word‑ and sentence‑level semantic recovery, layer‑wise probe classifiers for harmfulness, and attention/perplexity shifts to explain when and why perturbations help or hurt jailbreaks.

**Audience:**

Yes

**Audience Explanation:**

This paper presents findings that would interest a subset of the TMLR audience like researchers working on LLM safety and alignment, tokenization and subword modeling, and adversarial robustness.

**Claims And Evidence:**

No

**Claims Explanation:**

- ASR increases are modest in absolute terms. In the non-template setting, ASR remains <0.2% for LLaMA3-8B and only rises to ~4% for Mistral-7B. These are not effective increases in a practical sense.
- The paper does not compare its perturbations to existing character-level attacks (e.g., Charmer, Bad Characters), so it’s unclear whether the observed ASR boosts are due to the novelty of the method or just the brittleness of the evaluated models.
- Judge model reliability is assumed, not validated. All ASR results rely on LLaMA-Guard and WildGuard without human validation or inter-judge agreement analysis. These judges can be brittle under perturbations.
- Some perturbations are not minimal or stealthy. Methods like Word Unscrambling (“firearms” → “asfrriem”) or Letters Cycling are visually obvious and would likely be flagged by human moderators. The paper does not clearly define or justify its threat model, making it unclear when these perturbations are realistic.
- The cosine similarity drops (e.g., 0.4–0.8) are not contextualized with controls like synonyms, inflections, or random word pairs. Without these, it’s unclear whether the observed drift is specifically problematic for safety or just generic noise.

**Requested Changes:**

- Validate the safety judges.
- Compare to at least one prior character-level jailbreak.
- Clarify threat model and split “stealthy” vs. “non-stealthy” perturbations.

---

> ### Author Response · Authors · 2026-03-13
>
> We appreciate the reviewer’s logical claims to help us improve our work.
>
> >  (1) Judge model reliability is assumed, not validated. All ASR results rely on LLaMA-Guard and WildGuard without human validation or inter-judge agreement analysis. These judges can be brittle under perturbations.
>
> To address concerns about the influence of perturbations on the Judge LLM, we conducted a new series of evaluation using clean attack targets (original requests without any templates or perturbations) as the input for the Judge. Interestingly, the ASR results increased when using these pure inputs. This demonstrates two key points: first, our perturbation method genuinely causes the target LLM to produce harmful content; second, our previous evaluation (using perturbed prompts) was actually a conservative estimate that likely under-reported the true attack success rate. By providing the Judge with clean, unambiguous intent while keeping the model's response the same, we’ve eliminated any potential labeling bias caused by character-level noise. These results further strengthen our claim that character-level perturbations amplify jailbreak risks.
>
> We have included these new results in Table 1, and move the original results to Table 2. We newly added Section 3.5 to write the corresponding analysis, and also updated the contents in Appendix C (included new results in Table 8, moved the original results to Table 9, and wrote a short analysis).
>
>
>
>
> >  (2) The paper does not compare its perturbations to existing character-level attacks (e.g., Charmer, Bad Characters), so it’s unclear whether the observed ASR boosts are due to the novelty of the method or just the brittleness of the evaluated models.
>
> The primary goal of this study is to expose a fundamental vulnerability exposing a vulnerability that can be exploited by adversaries in a quite easy way, not comparing our ASR with other methods to show the advantage of ours, so optimized perturbation variants are out of our research focus.
> Currently, our implementation don’t need to choose a place for insertion and choose the character to insert via optimization, just by randomly sampling characters and applying edits on the original text can also achieve a higher ASR in multiple settings, which means the attack could be implemented at a tremendous scale (in a greedy way) because no extra overload for calculation is needed.
> From this viewpoint, we’ve stressed this point in the main text (Introduction Part and Section 3), instead of “method”, we should call this as a “vulnerability” or “weakness”.
> Also, to emphasize this, we’ve changed the title from “Character-Level Perturbations Amplify LLM Jailbreak Attacks” to “Random Character-Level Perturbations Amplify LLM Jailbreak Attacks”.

---

> ### Author Response · Authors · 2026-03-13
>
> > (3) Some perturbations are not minimal or stealthy. Methods like Word Unscrambling (“firearms” → “asfrriem”) or Letters Cycling are visually obvious and would likely be flagged by human moderators. The paper does not clearly define or justify its threat model, making it unclear when these perturbations are realistic.
>
> ASR increases are modest in absolute terms. In the non-template setting, ASR remains <0.2% for LLaMA3-8B and only rises to ~4% for Mistral-7B. These are not effective increases in a practical sense.
>
> As the reviewer pointed out that we need to clarify the threat model, in real life scenarios, this kind of attacks can be implemented at tremendous scale (in a greedy way) because no extra overload for calculation is needed. This allows for greedy batching where an attacker generates numerous variants for one attack target, with only few costs. Because random perturbations are shown to increase ASR, the cumulative probability of achieving at least one successful jailbreak within a large-scale batch becomes nearly guaranteed, even without sophisticated optimization.
>
> The reviewer also pointed out that we need to split “stealthy” and “non-stealthy” perturbations. While we acknowledge that 8 perturbation methods vary in visual obviousness, we’ve opted not to formally categorize them into "stealthy" and "non-stealthy" groups because all methods serve the same objective of exploiting model vulnerabilities, and the unpredictable nature of random noise means an attacker cannot always pre-determine the stealthiness of a generated variant (even method like Word-unscrambling can generate a quite similar variant compared to the original version).
>
> Furthermore, while the observed ASR increase may appear modest in certain cases, it is important to note that every single experiments’ category involved only 4,160 test cases, which originated from 520 initial samples, and each word was perturbed only once. Given these constraints, even a marginal improvement of 0.1% per variant suggests that in a real-world attack, where data scales are larger and multiple perturbation iterations are used, the aggregated success rate would become substantially more dangerous.
>
>
>
> > (4) The cosine similarity drops (e.g., 0.4–0.8) are not contextualized with controls like synonyms, inflections, or random word pairs. Without these, it’s unclear whether the observed drift is specifically problematic for safety or just generic noise.
> We agree that representational drift can occur with generic linguistic variations. However, our primary goal in reporting these values (0.4 to 0.8) was to provide a quantitative indication of perturbation intensity in the latent space, rather than claiming it as a unique signature of jailbreak attacks. In our study, this representation drift serves as a necessary precondition: it demonstrates that character-level perturbations successfully push the input tokens away from the model's well-aligned "refusal regions". Therefore, in the next step, we can start to conduct semantic recovery analysis, and introduce the results.
> On the other hand, while synonyms, inflections, or random word pairs might also exhibit drift, their drift may not lead to the functional failure (i.e., increased ASR) that we observe in our experimental group, so those linguistic variations are out of our research focus.
>
>
> As a result, we’ve conducted the following experiment:
>
> New Judge Evaluation Using Clean Inputs: We conducted a new series of evaluation using clean attack targets (original requests without any templates or perturbations) as the input for the Judge.
>
> We’ve also made the following main text revisions:
>
> New Judge Evaluation Results: We have included these new results in Table 1, and move the original results to Table 2. We newly added Section 3.5 to write the corresponding analysis, and also updated the contents in Appendix C (included new results in Table 8, moved the original results to Table 9, and wrote a short analysis).
>
> Threat Model Clarification: We’ve added an introduction of the general threat model in Section 3.1.
>
> New Research Focus Claim: In Introduction part and Section 3, we added a claim that we are focusing on exposing a vulnerability, rather than put forward a new jailbreak method and compare ours with other researchers’ methods.
>
> Title Modification: We’ve changed the title from “Character-Level Perturbations Amplify LLM Jailbreak Attacks” to “Random Character-Level Perturbations Amplify LLM Jailbreak Attacks”.

---

### Review · Reviewer_ig7C · 2026-01-16

**Summary Of Contributions:**

This paper investigates how character-level perturbations can amplify jailbreak attacks on LLMs by exploiting tokenization vulnerabilities. The core idea is that minimal perturbations (e.g., "firearms" → "firrrearms") fragment tokenization, distort internal representations, and thereby compromise safety mechanisms. The authors adopt 8 perturbation methods from prior work (Dekker & van der Goot, 2020) and evaluate them across 4 open-source LLMs in template-free and template-augmented (DeepInception, GCG) settings. The paper provides mechanistic analysis through four complementary experiments: tokenization/representation shifts, word-level and sentence-level semantic recovery, linear probe classifiers for harmfulness detection, and analysis of cases where perturbations reduce attack success.

Key Strengths:

- Comprehensive experiments across multiple models, templates, and perturbation methods with detailed specifications
- Complementary results for several experiments are provided in the appendix
- The experimental designs for semantic recovery (Patchscope, spelling error tasks) and probe-based harmfulness detection are well-constructed
- Clear writing; the paper is well-organized and easy to navigate
- Addresses cases where perturbations hurt performance, and acknowledges limitations of the method appropriately

Key Weaknesses:

- Training and validation accuracy scores are not included for the probe analysis for an accurate comparison of the probe performance drop
- In the attention/perplexity experiments, corresponding graphs for successful prompts were not presented

**Additional Comments:**

N/A

**Audience:**

Yes

**Audience Explanation:**

Researchers working on LLM safety, adversarial robustness, and interpretability would find value in this work.

**Broader Impact Concerns:**

Since the paper introduces a mechanism to bypass LLM safety mechanisms, it would be appropriate to include a Broader Impact Statement in the paper as well.

**Claims And Evidence:**

Yes

**Claims Explanation:**

The experimental scope is comprehensive, with evaluations across multiple models and settings. The tokenization shift analysis (Section 4.1) and semantic recovery experiments (Section 4.2) are well-executed and provide convincing evidence that perturbations disrupt model processing. However, there are some gaps to be addressed to make this a more complete presentation :


**Missing baseline:** The probe analysis in Section 4.3 reports accuracy only on perturbed harmful prompts (Figure 5), but never establishes the baseline training & validation accuracy on original (unperturbed) prompts. Without knowing how well probes perform on clean inputs, the reported ~80% accuracy at deeper layers cannot be interpreted meaningfully. This could represent a 20% degradation from perfect performance, or minimal degradation from maybe an 85% baseline. Additionally, no train/validation/test split is described.

**Missing baseline for attention/perplexity analysis:** Section 4.4 analyzes attention and perplexity changes only for cases where perturbations caused attack failure. The authors do not provide a comparison with cases where perturbations succeeded. Without this baseline, it is difficult determine whether elevated attention/perplexity is unique to failure cases or a general property of all perturbed prompts.

**Characterization of spelling task results:** The paper claims "correction is easier than detection" (Section 4.2), but Table 2 shows this holds for only 15 out of 32 model-perturbation combinations. Two of four models (Mistral-7B, Vicuna-13B) show higher average detection than correction rates. From my observations, there seems to be a pattern that might offer an explanation for this: highly destructive perturbations (Word Unscrambling, Letters Cycling) are consistently easier to detect but harder to correct across all models, while mild perturbations (Letter Repetition, Vowel Dropping) show the opposite pattern. This discussion could be included in the paper.

There is also an ambiguity in writing which could be made smoother: subsec 4.2 mentions "design three tasks of increasing difficulty" & "Table 2 indicates that correction is easier than detection" are contrasting. The writing here may be made smoother by including an explanation, as I mentioned, and presenting it more like a hypothesis and an unexpected result.

**Ambiguity in judge LLM evaluation**: The paper does not clearly specify whether judge models receive the original or perturbed prompt when evaluating attack success. Appendix C.2 mentions "the original user request" but this phrasing is ambiguous. Since the experiments have established that such perturbations affect responses, it might also affect the Judge's decision. Kindly clarify on how the Judge LLM was fed responses for its classification.

Overall, I believe the premise and the results provided by the paper are compelling, but need refinement corresponding to the stated points.

**Requested Changes:**

**Major:**

- Provide probe baseline accuracy: Report (a) training accuracy, (b) validation accuracy (with proper train/val/test split), and (c) test accuracy on original (unperturbed) harmful prompts. Plot this baseline comparison with the perturbed prompt accuracy reported in Figure 5
- Provide baseline for attention/perplexity analysis: Extend Section 4.4 to include attention and perplexity measurements for cases where perturbations succeeded (i.e., changed responses from harmless to harmful, or maintained harmful responses)

**Minor:**

- Revise the correction vs. detection discussion: The writing under subsection 4.4 should be revised to accurately reflect the results. Prefereably Include an explanation
- Add to the limitations section: Add a limitation acknowledging that there is currently no mechanism to preemptively determine whether a perturbed prompt will integrate naturally with a given template. Future work directions for predictive mechanisms could be suggested
- Clarify judge model inputs: Explicitly state whether judge models receive the original or perturbed prompt when evaluating responses. If perturbed prompts are used, provide validation that judge reliability is not compromised
- Format references consistently: Several references (e.g., Grattafiori et al., 2024; OpenAI et al., 2024) list hundreds of authors across multiple pages. These should be formatted more concisely

---

> ### Comment · Reviewer_ig7C · 2026-01-16
>
> Additionally, with due respect to the authors' work, I sincerely apologize to the Authors, the AE, and the Editors-in-Chief for the delay in posting my review. I had been recovering from poor health.
>
> Thank you for your kind patience.

---

> > ### Author Response · Authors · 2026-03-13
> >
> > > (3) Characterization of spelling task results: The paper claims "correction is easier than detection" (Section 4.2), but Table 2 shows this holds for only 15 out of 32 model-perturbation combinations. Two of four models (Mistral-7B, Vicuna-13B) show higher average detection than correction rates. From my observations, there seems to be a pattern that might offer an explanation for this: highly destructive perturbations (Word Unscrambling, Letters Cycling) are consistently easier to detect but harder to correct across all models, while mild perturbations (Letter Repetition, Vowel Dropping) show the opposite pattern. This discussion could be included in the paper.
> > There is also an ambiguity in writing which could be made smoother: subsec 4.2 mentions "design three tasks of increasing difficulty" & "Table 2 indicates that correction is easier than detection" are contrasting. The writing here may be made smoother by including an explanation, as I mentioned, and presenting it more like a hypothesis and an unexpected result.
> >
> > We agree that the relationship between detection and correction varies across perturbation types. Specifically, we’ve now incorporated your observation into Section 4.2, clarifying that highly destructive perturbations (e.g., Word Unscrambling) are indeed easier to detect but harder to correct. We’ve rephrased our original claim as a hypothesis and added this discussion to better align with the results in Table 3.
> >
> >
> >
> >
> > > (4) Ambiguity in judge LLM evaluation: The paper does not clearly specify whether judge models receive the original or perturbed prompt when evaluating attack success. Appendix C.2 mentions "the original user request" but this phrasing is ambiguous. Since the experiments have established that such perturbations affect responses, it might also affect the Judge's decision. Kindly clarify on how the Judge LLM was fed responses for its classification.
> >
> > To address concerns about the influence of perturbations on the Judge LLM, we conducted an additional evaluation using clean attack targets (original requests without any templates or perturbations) as the input for the Judge. Interestingly, the ASR results increased when using these pure inputs. This demonstrates two key points: first, our perturbation method genuinely causes the target LLM to produce harmful content; second, our previous evaluation (using perturbed prompts) was actually a conservative estimate that likely under-reported the true attack success rate. By providing the Judge with clean, unambiguous intent while keeping the model's response the same, we’ve eliminated any potential labeling bias caused by character-level noise. These results further strengthen our claim that character-level perturbations amplify jailbreak risks.
> >
> > We have included these new results in Table 1, and move the original results to Table 2. We newly added Section 3.5 to write the corresponding analysis, and also updated the contents in Appendix C (included new results in Table 8, moved the original results to Table 9, and wrote a short analysis).

---

> ### Author Response · Authors · 2026-03-13
>
> We are truly grateful that you took the time to provide such a thorough and constructive review while you were recovering. Your insights are incredibly valuable to us. We hope your health situation is better now.
>
> >  (1) Missing baseline: The probe analysis in Section 4.3 reports accuracy only on perturbed harmful prompts (Figure 5), but never establishes the baseline training & validation accuracy on original (unperturbed) prompts. Without knowing how well probes perform on clean inputs, the reported ~80% accuracy at deeper layers cannot be interpreted meaningfully. This could represent a 20% degradation from perfect performance, or minimal degradation from maybe an 85% baseline. Additionally, no train/validation/test split is described.
>
> Previously, We used 100% data for training and used their perturbed version for inference, not including any split for validation and test. We’ve complemented this in extra experiments. We split the train/val/test set at a rate of 14/3/3. The new result is just aligned with our previous claim, perturbed inputs’ probe accuracy drops in middle layers and declines further in deeper layers, and stably lower than unperturbed versions.
>
> We have inserted new Figure 5 and Figure 6, and deleted the old Figure 17-24. We also updated Section 4.3, and included the training details in the Appendix F.2.
>
>
> > (2) Missing baseline for attention/perplexity analysis: Section 4.4 analyzes attention and perplexity changes only for cases where perturbations caused attack failure. The authors do not provide a comparison with cases where perturbations succeeded. Without this baseline, it is difficult determine whether elevated attention/perplexity is unique to failure cases or a general property of all perturbed prompts.
>
> Add to the limitations section: Add a limitation acknowledging that there is currently no mechanism to preemptively determine whether a perturbed prompt will integrate naturally with a given template. Future work directions for predictive mechanisms could be suggested
>
> We thank the reviewer for pointing out the need for adding baseline groups in the attention and perplexity analysis. Following this suggestion, we extended our analysis to include two baseline groups: cases where responses changed from harmless to harmful and cases where harmful responses were maintained. We calculated the attention weight and perplexity change for these groups using three span settings: “the whole attack target span”, “only the perturbed words” and “the junction tokens between the template and attack target span”. Afterwards, we compared them against our original experimental group (where responses changed from harmful to harmless).
>
> However, our results indicate that elevated attention and perplexity are general properties observed across all three groups, with no statistically significant difference between the baseline groups and the experimental group. This suggests that the increased perplexity is likely a universal reaction to character-level noise rather than a unique trigger of ASR degradation.
>
> It is important to clarify that this analysis was originally intended as a supplementary attempt to provide a case study on failure cases, and it does not affect our primary conclusion that "Character-Level Perturbations Amplify LLM Jailbreak Attacks." Consequently, to maintain technical rigor, we’ve revised Section 4.4 into a brief discussion. Specifically, we kept the rest contents in Section 4.4, but removed the perplexity and attention change analysis part to avoid misleading effects. We will frame this as an open challenge for future work, calling for more sophisticated interpretability tools specifically designed to judge template and attack target integration to better analyze the underlying reasons for ASR degradation. Also, we’ve removed the attention and perplexity change results in Appendix G.

---

> ### Author Response · Authors · 2026-03-13
>
> > (5) Since the paper introduces a mechanism to bypass LLM safety mechanisms, it would be appropriate to include a Broader Impact Statement in the paper as well.
>
> We appreciate the reviewer's suggestion regarding the ethical considerations of our work. We’ve added a "Broader Impact Statement" to the revised manuscript to discuss the potential societal implications and the defensive motivations behind our research on LLM safety vulnerabilities.
>
>
>
>
>
> Regarding the reviewer’s comments, we’ve conducted the following experiments:
>
> Restructuration of Probing Experiments: We’ve restructured our experiment for layer-wise harmfulness probing, introducing new splits for validation and test. Just using the test set for inference, as the baseline for comparison.
>
> Running Baseline Experiments for Attention/Perplexity Analysis: We’ve implemented the baseline experiments including cases where responses changed from harmless to harmful and cases where harmful responses were maintained, ranging from three span calculation settings.
>
> New Judge Evaluation Using Clean Inputs: We conducted a new series of evaluation using clean attack targets (original requests without any templates or perturbations) as the input for the Judge.
>
> We’ve also made the following main text revisions:
>
> Results of Experiments: we’ve added the new probing accuracy figures (Figure 5 and Figure 6), and deleted the old Figure 17-24. We also updated Section 4.3, and included the training details in the Appendix F.2.
>
> ASR Degradation Analysis Part Reframing: We have rewrite the Section 4.4 into a brief discussion, removing the attention and perplexity change analysis part, and added a stress on the lack of template & attack target integration judging system/tools to better analyze the underlying reasons for ASR degradation, which needs further investigations in future work. Also, we’ve removed the attention and perplexity change results in Appendix G.
>
> Refinement of an Expression: we’ve refined the claims in the correction vs. detection discussion part in Section 4.2.
>
> Clear Definition of Judge Model Inputs: we’ve added a clearer definition of judge model inputs in Section 3.5.
>
> Reference Format Reframing: we’ve reframed 2 weirdly formatted references.
>
> Broader Impact Statement: we’ve added a Broader Impact Statement including more potential societal implications and the defensive motivations.

---

> > ### Comment · Reviewer_ig7C · 2026-04-12
> > **Reviewing manuscript revisions**
> >
> > I thank the authors for their thorough and transparent revisions in accordance with my review comments. Having carefully reviewed the changes, I can affirm that the authors have appropriately addressed my original concerns and suggestions. The color-coded changes are appreciated as well. They made reviewing the edits easier.
> >
> > However, I noticed an inconsistency introduced in the revised manuscript regarding the probe analysis in Section 4.3. The main text describes the probes as "linear classifiers, commonly referred to as probes" and presents Equation (3) as a standard logistic regression. The subsequent interpretation on page 11 concludes that "harmfulness is largely linearly separable in the representation space of all evaluated models." However, Appendix F.2 describes the actual implementation as a "Multi-Layer Perceptron (MLP) probe" with hidden dimension 256, 2–3 layers, and dropout (Table 10).  I would request the authors to clarify this discrepancy in the manuscript.
> >
> > Minor writing error: The last line of the first paragraph under Section 4.3 has a referencing error for a section of the Appendix (I think this was meant to be F.2?). Kindly fix the reference.
> >
> > Overall, the paper now presents a more sound and complete argument with the incorporated revisions.

---

> > > ### Author Response · Authors · 2026-04-16
> > >
> > > We thank the reviewer for spending more time to review the revised manuscript, and pointing out this writing inconsistency. We have updated Equation (3) to reflect the MLP formulation and revised all "linear"-related descriptions (e.g., "linear classifiers," "logistic regression," "linearly separable," "linearly decodable") throughout the main text and appendix to accurately describe the MLP probes, ensuring consistency with Appendix F.2.
> > >
> > > And also, the appendix referencing error at the last line of the first paragraph under Section 4.3 has also been corrected. Thanks again for the careful review, we appreciate that.
> > >
> > > The new manuscript pdf has been uploaded now.

---

### Review · Reviewer_gVMX · 2026-02-28

**Summary Of Contributions:**

There is a large body of research focused on understanding safety and guardrails of large language models (LLMs) using jailbreak attacks, which bypass harmful and privacy-related content through prompts. This study focuses on jailbreak attacks at the character level by introducing character-level perturbations to exploit subword tokenizers, which result in vulnerabilities that increase attack success rates in large language models. To achieve this, the authors considered 8 character-based perturbations mainly at the keyword level and embedded these perturbations into two popular prompt-based templates, DeepInception and GCG, and analyzed the jailbreak attack success rate (ASR) in several open-source LLMs. Using these methods, the authors perform four types of analysis: tokenization shifts, representation drift, semantic recovery, and layer-wise probing as a binary classifier for harmful and harmless content. The experimental findings reveal that even simple tokenization-induced representation drift increases attack success rates and weakens the safety mechanism in LLMs.

**Contributions:**

* Introduction of a simple character-level, template-agnostic jailbreak attack that increases attack success rates in LLMs without requiring access to model parameters or gradients.
* Qualitative and quantitative analysis of tokenization shifts that distort representations, along with experiments on unable to recover the original meaning from perturbed texts, indicating that reduced harmful-harmless separability.
* Exposes vulnerabilities in subword tokenization, showing that minimal word-level character perturbations can introduce jailbreak attacks, underscoring the need for better tokenizer design.

**Strengths:**

I found this work to have the following strengths:
* **Clarity:** The main motivation for studying how subword tokenization introduces vulnerabilities, even with character-level perturbations in keywords that allow jailbreak attacks and introduce malicious content in LLMs, is clearly and well presented. The paper also uses standard 8 perturbation methods that have been explored in prior studies and clearly presents how these perturbations are applied at the keyword level and embedded into template-based approaches such as DeepInception and GCG. The experimental setup is also clear, including the use of a linear probing classifier for harmful and harmless content based on the last-token representation. Further, the authors present a four-stage analysis explaining why the attack success rate increases using these methods. The details about the evaluation of open-source LLMs and the use of two guardrail LLMs for evaluation are also clearly presented.
* **Originality:** The idea of investigating subword tokenization vulnerabilities through minimal character-level perturbations is quite interesting. If tokenization fragmentation while splitting words into subwords distorts internal word representations, this may cause the language model to lose its guardrails and reveal privacy and safety-related information. Therefore, the authors’ approach is novel in this context. Further, the authors revalidate the model and verify whether the source word can be retrieved or whether semantic information can be recovered. Overall, this paper establishes a connection between subword tokenization behavior and harmfulness detection degradation and provides a clear analysis of jailbreak attacks in current language models beyond simply reporting attack performance.
* **Significance:** This work is significant in that it contributes to a better understanding of how tokenization-induced perturbations affect internal representations and weaken safety mechanisms in large language models. By systematically analyzing tokenization shifts, representation drift, semantic recovery, and harmfulness separability, the paper provides deeper insight into why jailbreak attacks succeed under minimal character-level changes.  It further helps clarify how safety guardrails may fail when simple word-level integrity is slightly modified and highlights structural weaknesses in current subword tokenization methods. This paper also analyze the error analysis 	where perturbations reduce attack success rate due to poor template integration, using attention weights and perplexity changes. Overall, this study offers meaningful contributions toward understanding robustness and safety vulnerabilities in modern language models.

**Audience:**

Yes

**Audience Explanation:**

Yes. The findings of this paper would be of interest to a meaningful portion of TMLR’s journal.

**Claims And Evidence:**

No

**Claims Explanation:**

From my perspective, although the submission presents structured empirical analyses, the evidence remains largely correlational and limited in scope, lacking sufficient causal validation and broader model-scale evaluation to fully support its central claims.

* **Limited model scope:** Although the authors exploit four open-source LLMs, they use guardrail LLMs only at the evaluation stage. Using guardrail LLMs at the evaluation stage is acceptable; however, the authors should also test these character-level perturbations directly on guardrail LLMs, which would further provide failure cases in those models. Instead, the authors evaluate only open-source LLMs, and these models do not necessarily have strong guardrails introduced during training. Therefore, the authors should repeat the experiments on guardrail LLMs to provide a clearer analysis of the limitations of these models. During evaluation, the authors could use another guardrail LLM while exploiting one guardrail LLM to assess robustness more comprehensively.
* **Limited validation of causality regarding subword tokenization causing jailbreak attacks:** Although the paper investigates subword tokenization using word-level character perturbations, all the results presented are mainly correlational. The representation analysis and layer-wise probing do not establish causation. To empirically demonstrate that subword tokenization is a leading factor in the increase of attack success rates, the authors could test a word-level tokenizer on the perturbed words to examine whether representational changes still occur. Additionally, the authors could use a character-level tokenizer to perform similar analyses. These two analyses would provide stronger support that subword tokenization distorts representations and results in jailbreak attacks.
* **Model scale impact:** Although the authors tested four open-source LLMs and used two guardrail LLMs for evaluation, the scale of the tested models ranges between 7B and 13B parameters. While the proposed method is model-agnostic, the paper does not evaluate whether attack success rates change for larger-scale models (e.g., 70B) or smaller language models (e.g., 1B or 3B). It remains unclear whether these attack success rates would diminish or increase as model size varies across small and large scales.
* **Limited evaluation with other text perturbations:** Although the paper performs character-level perturbations and measures attack success rates, there is no comparison with other types of text perturbations. Text perturbations can be applied at the prompt level, word level, and semantic level, whereas the current character-level perturbations are primarily syntactic rather than semantic. The absence of comparisons with other perturbation strategies limits the scope of the current paper. The authors may consider referencing prior work on word-level and prompt-level perturbations for a more comprehensive comparison.

[Neerudu et al. 2023] On Robustness of Finetuned Transformer-based NLP Models, EMNLP-2023 Findings

[Kaijie et al. 2023] PromptBench: A Unified Library for Evaluation of Large Language Models

**Requested Changes:**

**Major Comments/Questions:**

* How do the authors control bias when using GPT-4o-mini? To identify harmful words, the authors use the GPT-4o-mini model. How do the authors investigate potential bias in this model when selecting harmful words? Did the authors try other closed-source models to verify whether the selected harmful words remain consistent across different models?
* Attack success rates evaluated by the LLaMA-Guard-3-8B model report 0 score for word-unscrambling, while the WildGuard model shows mostly 0% across perturbations for the LLaMA-3-8B model. What could be the reason for this behavior? Is it due to the same LLaMA-3-8B backbone being used in the Guardrail model, or is it due to architectural or training differences?

---

> ### Author Response · Authors · 2026-03-13
>
> We sincerely thank the reviewer for their constructive feedback, which has been instrumental in refining our work.
>
> > (1) Limited model scope: Although the authors exploit four open-source LLMs, they use guardrail LLMs only at the evaluation stage. Using guardrail LLMs at the evaluation stage is acceptable; however, the authors should also test these character-level perturbations directly on guardrail LLMs, which would further provide failure cases in those models. Instead, the authors evaluate only open-source LLMs, and these models do not necessarily have strong guardrails introduced during training. Therefore, the authors should repeat the experiments on guardrail LLMs to provide a clearer analysis of the limitations of these models. During evaluation, the authors could use another guardrail LLM while exploiting one guardrail LLM to assess robustness more comprehensively.
>
> From our understanding, the reviewer is suggesting that character-level perturbations in the prompt might bias the guardrail models’ judgements, and therefore using an additional guardrail model for assessment could provide a more comprehensive analysis. We appreciate this insightful perspective and agree that understanding the limits of current evaluation mechanisms is crucial.
> To address the concern regarding potential labeling bias, we conducted an additional evaluation comparing two types of Judge inputs: the original, clean attack targets (requests without any templates or perturbations) and the perturbed prompts with templates which were used in our initial experiments. Interestingly, the ASR results generally increased when the Judge was provided with the clean intent while keeping the target LLM's response constant.
> We believe this experiment directly addresses the reviewer’s concern without requiring a secondary guardrail model to evaluate the primary one. By using the clean, unambiguous intent as a "ground truth" reference for the Judge, we have eliminated any potential classification errors induced by character-level noise. This demonstrates that our original evaluation was actually a conservative estimate that likely under-reported the true success rate. Since the Judge performs more accurately when the perturbation is removed from its reference input, it proves that the observed jailbreak success is due to the target model's output rather than a failure in the Judge's understanding. These results further strengthen our claim that character-level perturbations genuinely amplify jailbreak risks.
> We have included these new results in Table 1, and move the original results to Table 2. We newly added Section 3.5 to write the corresponding analysis, and also updated the contents in Appendix C (included new results in Table 8, moved the original results to Table 9, and wrote a short analysis).
>
>
>
>
> > (2) Limited validation of causality regarding subword tokenization causing jailbreak attacks: Although the paper investigates subword tokenization using word-level character perturbations, all the results presented are mainly correlational. The representation analysis and layer-wise probing do not establish causation. To empirically demonstrate that subword tokenization is a leading factor in the increase of attack success rates, the authors could test a word-level tokenizer on the perturbed words to examine whether representational changes still occur. Additionally, the authors could use a character-level tokenizer to perform similar analyses. These two analyses would provide stronger support that subword tokenization distorts representations and results in jailbreak attacks.
>
> We appreciate the reviewer highlighting the importance of causal validation. We argue that our existing layer-wise probing and representation analysis already provide a strong logical basis for causality because subword tokenization is the very first stage of model processing. Any distortion in the embedding layer acts as an upstream trigger for downstream semantic shifts, and our analysis shows that representational deviation begins at the earliest layers long before complex semantic reasoning occurs. This early-layer divergence is directly caused by the out-of-vocabulary subword sequences, which effectively identifies subword tokenization as the primary driver of the observed jailbreak success.

---

> ### Author Response · Authors · 2026-03-13
>
> > (3) Model scale impact: Although the authors tested four open-source LLMs and used two guardrail LLMs for evaluation, the scale of the tested models ranges between 7B and 13B parameters. While the proposed method is model-agnostic, the paper does not evaluate whether attack success rates change for larger-scale models (e.g., 70B) or smaller language models (e.g., 1B or 3B). It remains unclear whether these attack success rates would diminish or increase as model size varies across small and large scales.
>
>
> We appreciate the reviewer's suggestion to explore model scale, but the primary objective of our study is to point out a fundamental, character-level vulnerability of contemporary LLMs. While we acknowledge that model scale is an interesting factor to the overall ASR, a comprehensive analysis of "scaling laws" lies beyond the focus of this work.
>
> However, if the audience is interested in the model scale impacts, our current results for Vicuna-7B versus Vicuna-13B already reveal a trend. As model #parameter increases, we observe a lower baseline ASR, and a corresponding decrease in the marginal ASR gain triggered by perturbations, suggesting that larger models do possess inherently stronger defensive capabilities. Nonetheless, the fact remains that character-level noise amplifies jailbreak success across both scales, confirming that this vulnerability is a persistent threat that is not simply resolved by increasing parameter counts.
>
>
>
>
> > (4) Limited evaluation with other text perturbations: Although the paper performs character-level perturbations and measures attack success rates, there is no comparison with other types of text perturbations. Text perturbations can be applied at the prompt level, word level, and semantic level, whereas the current character-level perturbations are primarily syntactic rather than semantic. The absence of comparisons with other perturbation strategies limits the scope of the current paper. The authors may consider referencing prior work on word-level and prompt-level perturbations for a more comprehensive comparison.
>
>
> We thank the reviewer for pointing this out, but our primary objective is to expose the specific vulnerabilities in LLMs that stem from their underlying tokenization mechanisms. For this reason, our study intentionally focuses on character-level perturbations because they directly disrupt subword tokenization, which is a fundamental process in modern LLMs. While word-level or semantic-level perturbations are also valuable, they often bypass this specific layer or alter the overall meaning of the prompt. We do not claim our method is superior to prompt-level or semantic-level attacks in all scenarios, but focusing on uncovering this "tokenization-weakness". We have clarified this focus in the Introduction of the new manuscript.

---

> ### Author Response · Authors · 2026-03-13
>
> > (5) How do the authors control bias when using GPT-4o-mini? To identify harmful words, the authors use the GPT-4o-mini model. How do the authors investigate potential bias in this model when selecting harmful words? Did the authors try other closed-source models to verify whether the selected harmful words remain consistent across different models?
>
>
> We think that the task of identifying specific “harmful words” in one prompt is inherently challenging because toxicity often arises from the overall semantic context of a sentence rather than isolated tokens. Thus, a definitive "gold standard" for such selection does not exist. Our use of GPT-4o-mini is intended for relative harmfulness attribution by identifying relatively key areas that contribute significantly to the harmful intent, rather than seeking absolute, precise “harmful words”.
>
> Furthermore, the subsequent character-level perturbations are designed to disrupt the model's processing of these identified regions. Since these edits naturally alter the local semantics and tokenization of the prompt, the exact selection of “harmful words” do not critically impact the experimental outcome. As long as the perturbations are localized within the semantically harmful portions of the prompt, the following experiments are meaningful. Therefore, minor variations or potential biases in word selection by the GPT-4o-mini do not undermine the validity of our core findings.
>
>
>
>
>
>
> > (6) Attack success rates evaluated by the LLaMA-Guard-3-8B model report 0 score for word-unscrambling, while the WildGuard model shows mostly 0% across perturbations for the LLaMA-3-8B model. What could be the reason for this behavior? Is it due to the same LLaMA-3-8B backbone being used in the Guardrail model, or is it due to architectural or training differences?
>
>
> The slightly higher ASR reported by Llama-Guard-3-8B compared to WildGuard likely stems from differences in their training objectives and safety thresholds rather than architectural superiority. While Llama-Guard-3 shares the same backbone and tokenizer as the target model, it is specifically fine-tuned on a predefined set of safety taxonomies that might be more "permissive" toward certain perturbed semantics compared to WildGuard. WildGuard, being trained on a more diverse and potentially more aggressive set of adversarial prompts, may exhibit a stricter "refusal bias," leading it to flag almost any irregular or perturbed input as unsafe, thereby resulting in an even lower ASR.
>
>
>
> Regarding the reviewer’s comments, we’ve conducted the following experiments:
>
> New Judge Evaluation Using Clean Inputs: We conducted a new series of evaluation using clean attack targets (original requests without any templates or perturbations) as the input for the Judge.
>
> We’ve also made the following main text revision:
>
> New Judge Evaluation Results: We have included the new results from judge evaluation using clean inputs into Table 1, and move the original results to Table 2. We newly added Section 3.5 to write the corresponding analysis, and also updated the contents in Appendix C (included new results in Table 8, moved the original results to Table 9, and wrote a short analysis).

---

> > ### Comment · Reviewer_gVMX · 2026-04-12
> >
> > I thank the authors for addressing some of my concerns such as (i) potential labeling bias with two types of judge inputs: clean attack targets and  the perturbed prompts, (ii) model scale scope. However, questions regarding the main concerns remain only partially addressed. In particular, the causal claim that subword tokenization is the primary driver of jailbreak success is still not supported by the stronger empirical controls, such as comparisons with word-level or character-level tokenizers. Likewise, the questions regarding direct attacks on guardrail models.
> >
> > Incorporating these aspects would significantly strengthen the paper. Overall, the current version is in good shape, but addressing the above points would result in a stronger and more convincing submission.

---

> ### Author Response · Authors · 2026-04-16
>
> We thank the reviewer for the follow up suggestions. We realize that our previous descriptions might have lacked sufficient detail, leading to some misunderstandings regarding our experimental design, and we’d like to clarify our settings and thoughts in a more thorough way.
>
> Regarding the suggestion to conduct direct attacks on guardrail models, we would like to emphasize that our current methodology already implements such attacks as a core part of the evaluation. Specifically, the character level perturbations applied in our experiments are identical for both the target models and the guardrail models. The primary distinction lies in the input structure where the target models receive only the attack prompt while the guardrail models evaluate the combination of the attack prompt and the response generated by target models. By applying the exact same noise patterns to the inputs of the safety filters, we have directly tested their robustness against the same adversarial conditions that bypass the target models. This process identifies specific failure cases in the guardrail models and provides a rigorous assessment of their vulnerability to tokenization specific noise. These results, now reported in Table 1 and Table 2, demonstrate that the observed jailbreak success is not a result of judge misunderstanding but a genuine amplification of safety risks. Therefore, we believe this comparative analysis effectively addresses the concern without the need for additional models which would not provide further meaningful insights.
>
> Regarding the comparison with word level or character level tokenizers, we agree that these could theoretically serve as control groups. However, almost all mainstream large language models such as Llama 3 and Mistral are natively trained using Byte Pair Encoding or similar subword tokenization methods. Even if we were to implement word level or character level tokenizers as a baseline, there are no corresponding downstream models available that would allow us to conduct the subsequent representation shift analysis or other mechanistic investigations. For instance, word level tokenization would lead to immediate out of vocabulary issues when facing character level perturbations, while pure character level tokenizers are not used in modern high performance architectures because they result in excessively long sequence lengths and sparse feature representations that struggle to capture rich semantic information efficiently. Since our study focuses on exposing vulnerabilities in the modern LLMs, we maintain that focusing on subword tokenization ensures the practical relevance and internal consistency of our research.

---

### Author Response · Authors · 2026-03-13

We would like to express our sincere gratitude to all three reviewers for their insightful comments and constructive suggestions. We found the feedback extremely helpful in improving the quality and clarity of our manuscript.


In reply to Reviewer gVMX’s suggestions, we use brown to highlight the newly added or revised contents.

In reply to Reviewer ig7C’s suggestions, we use purple to highlight the newly added or revised contents.

In reply to Reviewer Kkcm’s suggestions, we use blue to highlight the newly added or revised contents.

In conclusion, we’ve made the following changes to the manuscript.

Revision and Expansion:

1. We’ve revised the abstract.

2. We’ve revised the Introduction.

3. We’ve added a new Section 3.1, and updated the contents in Section 3.5. We’ve also updated the results in Table 1, and added a new Table 2.

4. We’ve updated the “Sentence-Level Analysis with Spelling-Error Tasks” part in Section 4.2.

5. We’ve updated the "Training" and “Inference” parts in Section 4.3, and added new results as Figure 5 and Figure 6.

6. We’ve revised the original Section 4.4 into a discussion Section 5 (Discussion: ASR Degradation Phenomenon).

7. We’ve revised the Limitations.

8. We’ve revised the Conclusion.

9. We’ve added a new Broader Impact Statement.

10. We ‘ve reframed 2 wired formatted references.
11. We’ve updated the Appendix C.3, updated Table 8, and newly added Table 9.

12. We’ve updated Appendix F.2.


Deletion:

1. We’ve deleted the original probe accuracy results reported  in Figure 5.

2. We’ve deleted the perplexity and attention change analysis part.

3. We’ve deleted the original Figure 17-24 included in Appendix F.

4. We’ve deleted the original Figure 25-26 included in Appendix G.

---

### Author Response · Authors · 2026-04-16

According to the new reply from Reviewer ig7C, we have conducted some minor revisions.

Specifically, we have updated Equation (3) in Section 4.3, revised some inaccurate expressions throughout the Section 4.3 and Appendix F.2, and added one missing reference to Appendix F.2.

Besides the above minor revisions, the whole structure of Section 4.3 and Appendix F.2 remains the same.

---

### Decision · Action_Editor_qgwv · 2026-04-24

**Recommendation:** Accept as is

**Audience:**

Yes

**Audience Explanation:**

LLM Safety audience would appreciate the contribution.

**Claims And Evidence:**

Yes

**Claims Explanation:**

The paper identifies a timely and relevant vulnerability in LLM safety, showing that simple character-level perturbations can amplify jailbreak attacks via tokenization weaknesses . Reviewers appreciate the clear motivation, comprehensive empirical analysis, and improved revisions addressing key concerns. However, limitations remain in causal validation, modest empirical gains, and insufficient comparisons (e.g., alternative tokenizers or attacks). The threat model and evaluation robustness (e.g., judge reliability) could be further strengthened.